# Environmental Sustainability of Heat Produced by Poplar Short-Rotation Coppice (SRC) Woody Biomass

**Giulio Sperandio \*** , **Alessandro Suardi** , **Andrea Acampora and Vincenzo Civitarese**

Consiglio per la Ricerca in Agricoltura e l'Analisi dell'Economia Agraria (CREA), Centro di Ricerca Ingegneria e Trasformazioni Agroalimentari, Via della Pascolare 16, 00015 Roma, Italy; alessandro.suardi@crea.gov.it (A.S.); andrea.acampora@crea.gov.it (A.A.); vincenzo.civitarese@crea.gov.it (V.C.)
\* Correspondence: giulio.sperandio@crea.gov.it

**Abstract:** As demonstrated for some time, the reduction of greenhouse gases in the atmosphere can also take place using agroforestry biomass. Short-rotation coppice (SRC) is one of the sources of woody biomass production. In our work, the supply of woody biomass was considered by examining four different cutting shifts (2, 3, 4 and 5 years) and, for each, the Global Warming Potential (GWP) was evaluated according to the IPCC 2007 method. Regarding the rotation cycle, four biomass collection systems characterized by different levels of mechanization were analyzed and compared. In this study, it was assumed that the biomass produced by the SRC plantations was burned in a 350 kWt biomass power plant to heat a public building. The environmental impact generated by the production of 1 GJ of thermal energy was assessed for each of the forest plants examined, considering the entire life cycle, from the field phase to the energy production. The results were compared with those obtained to produce the same amount of thermal energy from a diesel boiler. Comparing the two systems analyzed, it was shown that the production and use of wood biomass to obtain thermal energy can lead to a reduction in the Global Warming Potential of over 70% compared to the use of fossil fuel.

**Keywords:** biomass; poplar; SRC; thermal energy; life cycle assessment; GWP; wood energy supply chain

## 1. Introduction

Given the current need for a progressive reduction in the use of fossil fuels, which are mainly responsible for $CO_2$ emissions and pollution in the atmosphere, support for the use of woody biomass as an alternative source to produce thermal and electrical energy represents an important aspect in the discussion on the supply of energy from renewable sources. Overall, compared to total primary energy, bioenergy accounts for around 9.5%, or 70% of the renewable energy consumed [1]. In the future, bioenergy consumption is expected to grow to up to 30% of renewables due to its significant use mainly in heat generation and the transport sector [2]. The European Union supports and promotes actions aimed at achieving a more sustainable economic-energy and environmental system aimed at the progressive reduction of the use of fossil fuels in favor of renewable energy sources such as bioenergy [3].

As reported by many studies, the use of bioenergy can contribute to a significant improvement in environmental impacts compared to that produced by fossil fuels [4,5]. Furthermore, the wider diffusion of biofuels will lead to a substantial reduction in greenhouse gas emissions, eutrophication, pollution, acidification and depletion of the ozone layer, with a consequent reduction in damage to human health [6]. There are several sources of biomass that can be used to generate different final forms of bioenergy (thermal, electrical, liquid fuels and biogas). Among these, an interesting source of biomass is represented by short-rotation coppice (SRC) plantations, characterized by a high planting density, and made with fast-growing species, such as poplar, willow and eucalyptus. These



crops, although currently covering only a few tens of thousands of hectares in Europe, can still represent an interesting production option for the purposes of achieving the objectives set by the European Union in terms of improving future environmental conditions. In particular, SRC plantations can play an interesting role in energy chains developed on a small scale in local rural districts mainly to produce thermal energy. In these cases, when planning the activation of energy chains, it is very important and useful to carry out an assessment of sustainability not only in economical but also in terms of environmental impact generated using bioenergy. In recent years, the life cycle assessment (LCA) methodology has been mainly used to estimate the positive or negative environmental impacts of processes associated with the production and use of biofuels [7–11]. The LCA methodology, albeit with its limitations, if supported by an inventory of primary data that allows multi-criteria analysis on the different phases of the production chains, can represent a very useful tool to provide indications and allow comparisons based on the different externalities deriving from different scenarios.

The aim of this study is to assess the carbon footprint of small-scale self-consumption wood-energy chains for heat generation based on SRC poplar plantations. The analysis is developed following the LCA method, applied to different scenarios concerning harvesting logistics and plantation cutting cycles. The biomass obtained is used to produce thermal energy in a local 350 kWt biomass plant. The carbon footprint of the energy chain, which includes the production of biomass and its transformation in a boiler, is compared with that of a conventional diesel-based boiler to produce the same amount of thermal energy.

## 2. Materials and Methods

### 2.1. Study Area and Poplar SRC Plantations

The experimental field was located in the North-East of Rome, within the farm of the CREA Research Centre for Engineering and Agro-Food Processing of Monterotondo, Italy (42°6′2.63″ N; 12°37′37.36″ E). The SRC poplar plantation of reference for the evaluation of the environmental impact analysis model with the LCA method was planted in 2005 on a flat surface of a total of 4.5 ha on a clayey soil with low organic matter content and phosphorus [12]. Three poplar clones were used: AF2 (*Populus* × *canadensis* Moenech), AF6 (*Populus nigra* L. × *Populus* × *generosa* A. Henry) and Monviso (*Populus* × *generosa* A. Henry × *Populus nigra* L.) [13,14]. The plantation was in single rows, spaced 2.80 m, while the cuttings on the row were 0.5 m apart, obtaining a density of 7140 trees ha$^{-1}$. For the purposes of this study, two periods of the production cycle were applied, namely 16 and 15 years. With reference to the first period, the harvesting operations were considered every 2 and 4 years. For the 15-year period, however, the harvesting operations were carried out every 3 and 5 years. A total of four biomass harvesting systems were considered: two systems applied in the cutting shifts of 2 and 3 years and other two different systems in the cutting shifts of 4 and 5 years. The harvesting systems in the 2- and 3-year cutting cycle were a two-step tractor-based harvesting system (TBHS) and a forage-based harvesting system (FBHS). The TBHS uses different equipment to perform the tree felling, then the extraction of whole trees and then chipping at the landing site. FBHS is a one-stage harvesting system that produces fresh wood chips directly in the field, where the biomass is unloaded into a trailer alongside the harvester and transported to the landing site for storage (Figure 1a) [15]. The two options considered for the 4- and 5-year cutting shifts were represented on the one hand by a manual felling of the trees with a chainsaw, the extraction of whole trees with a tractor equipped with a winch and subsequent chipping with a forest chipper at the landing site (Chainsaw-Based Harvest System—CBHS), and, on the other hand, by a feller-buncher for felling (Figure 1b), a skidder with grapple for trees extraction and a chipper before using the wood chips in the boiler (Shear head-Based Harvesting System—SBHS). It was considered that the whole biomass produced was used in the biomass boiler within the farm. From the combination of the cutting cycles and the types of mechanization adopted for the collection of the biomass, eight case studies were considered. In Scheme 1, the field operations on the plantations and the harvesting options

considered, together with the operations regarding the boilers (biomass and diesel boilers), are reported.

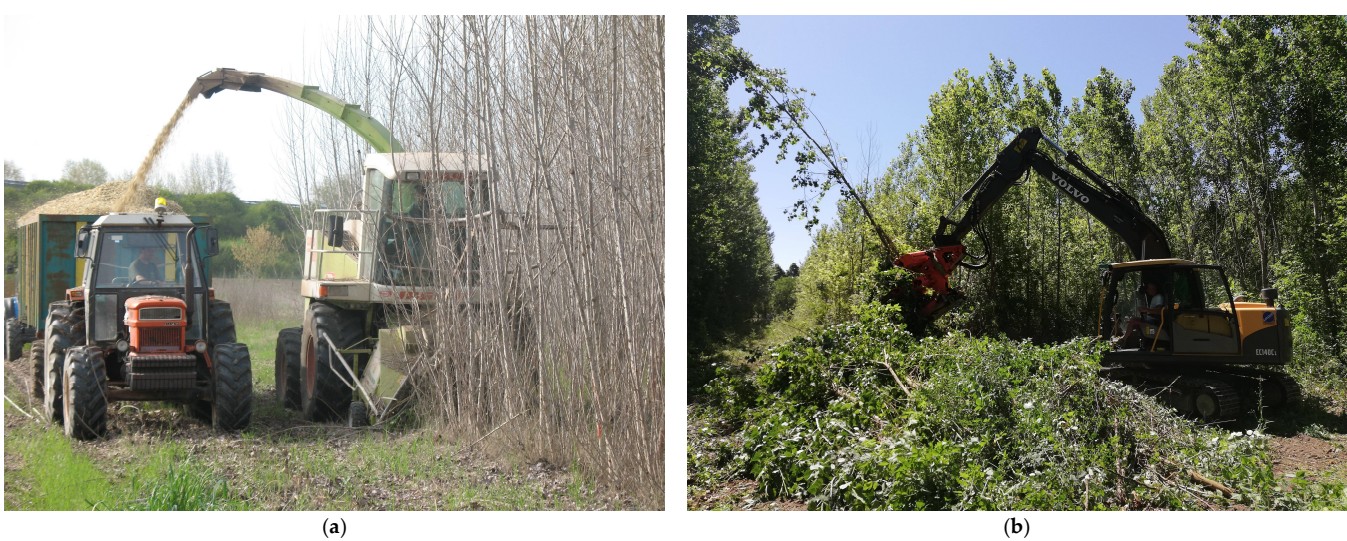

(**a**)　　　　　　　　　　　　　　　　　　(**b**)

**Figure 1.** Mechanized harvesting systems on poplar rotation coppice plantations: (**a**) Forage-Based Harvesting System (FBHS); (**b**) Shear head-Based Harvesting System (SBHS).

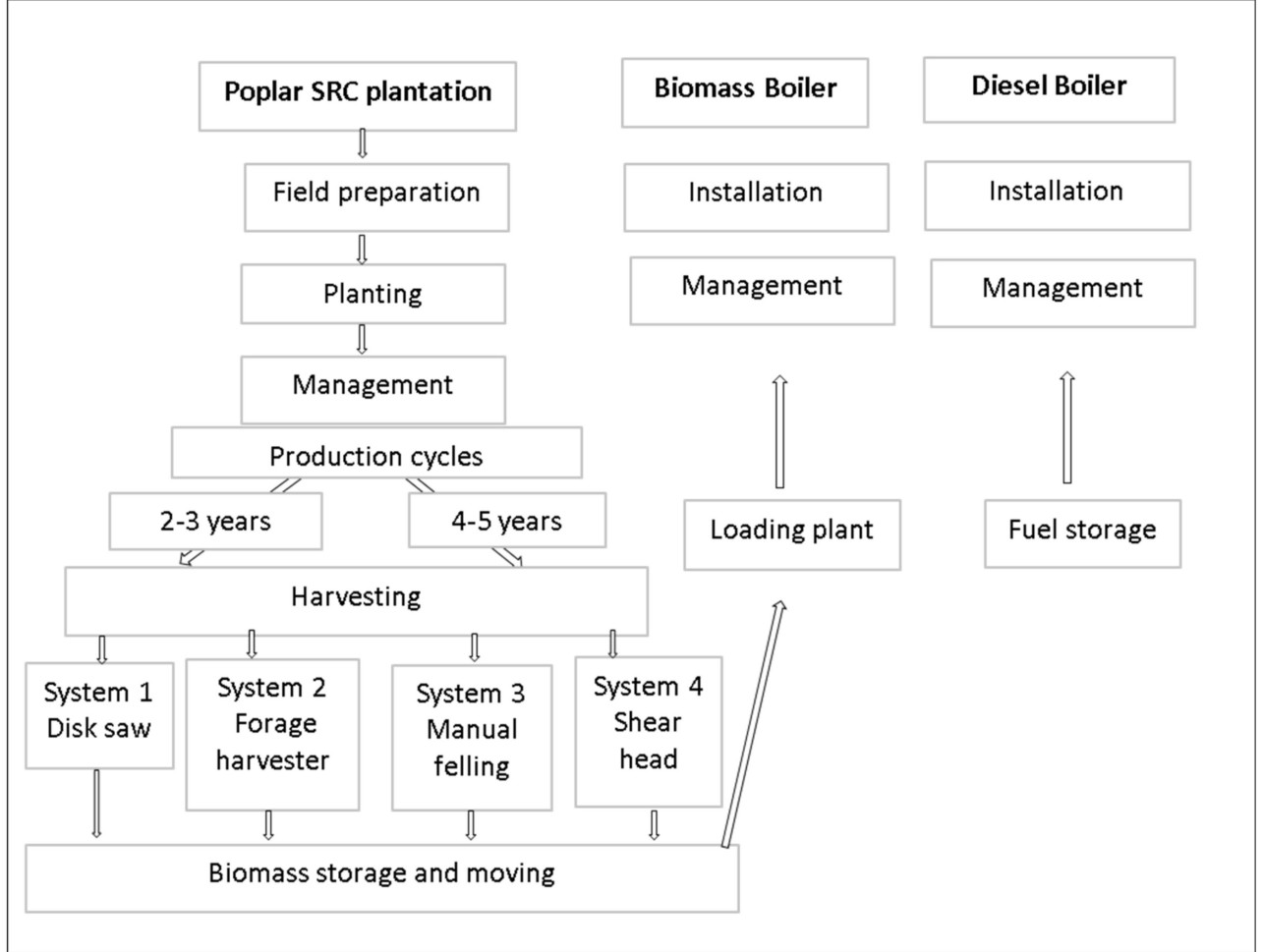

**Scheme 1.** Scheme of the poplar energy supply chain boundaries considered in the study.

### 2.2. Biomass and Diesel Boilers

The proposed environmental assessment model referred to the entire life cycle of the poplar plantation and the biomass power plant installed within the CREA farm. The biomass production energy system in the farm, and its use to produce thermal energy, was compared with a heating system of equivalent energy production but powered by fossil fuels (diesel).

This plant was used to heat the research center buildings, which were characterized by a potential volume to be heated of around 10,000 m$^3$. The biomass plant was equipped with a mobile grid, with a nominal power of 350 thermal kW. For the heating of the buildings, a period of 130 days per year was considered, calculating an average annual biomass supply of around 290 Mg, with a water content of 35%. For the purposes of this work, carbon footprint assessment generated by the thermal energy production of the biomass boiler was compared with that of the diesel boiler. The main parameters considered for the two boilers in comparison are shown in Table 1.

**Table 1.** Main parameters considered to evaluate the annual biomass or diesel consumption in the two boilers compared.

|  | Boilers | |
| --- | --- | --- |
|  | **Biomass** | **Diesel** |
| Building volume (m$^3$) | 9450 | 9450 |
| Operating period (days y$^{-1}$) | 130 | 130 |
| Heating period (h y$^{-1}$) | 3120 | 1560 |
| Rated thermal power (kWt) | 350 | 315 |
| Thermal efficiency of the boiler (%) | 81% | 90% |
| Lower heating value (LHV) (kWh kg$^{-1}$) | 3.11 | 11.86 |
| Water content (%) | 35.00% | ≤0.05% |
| Average biomass/diesel consumption (Mg y$^{-1}$) | 290.1 | 41.4 |

### 2.3. Environmental Analysis

The study evaluated the quantity of greenhouse gases emitted by poplar short- and medium-rotation coppices to produce thermal energy according to the LCA methodology. LCA is an in-depth "cradle-to-grave" analysis of the environmental impact of products or processes, and for this study, the impact category considered was the 100-year time horizon Global Warming Potential (GWP) based on the Intergovernmental Panel on Climate Change (IPCC) Fourth Assessment Report (AR4), published in 2007 [16]. In Table 2, for the eight different scenarios examined, the technical elements and the inputs used in the life cycle of the poplar plantations are reported. The CO$_2$ equivalent emissions per unit of thermal energy produced (1 GJ) downstream of each scenario were compared. The system evaluated the impact generated to produce 1 GJ of equivalent thermal energy from the agricultural, transport and transformation processes along the life cycle of the poplar groves, with reference to each cutting cycle and harvesting system considered. The functional unit was chosen to guarantee the comparison of the results obtained with other energy production systems, such as that from fossil sources. In the case of a small supply chain, the impact deriving from the production of 1 GJ of thermal energy produced by poplar wood chips in the biomass boiler was compared with 1 GJ produced by a diesel boiler. The system boundaries, i.e., the process units included in the LCA study, involved all the agricultural phases, the subsequent transport and transformation processes.

**Table 2.** Principal technical elements considered in the poplar plantations life cycle.

| Operation | Period (Years) | Machine | | | | | | Equipment | | Technical Input | | |
|---|---|---|---|---|---|---|---|---|---|---|---|---|
| | | Operation (n./ha) | Power (kW) | Weight (kg) | Work Time (h/ha) | Fuel (L/ha) | Machine (N.) | Weight (kg) | Type | Type | Quantity (kg) | Rates (kg/ha) |
| **Field preparation, planting and management** | | | | | | | | | | | | |
| - Deep scarification | 1 | 1 | 199 | 8700 | 3.50 | 136 | 1 | 800 | Ripper | | | |
| - Light ploughing | 1 | 1 | 199 | 8700 | 1.60 | 67 | 1 | 1100 | Plowshares | | | |
| - Fertilization (pre- and post-planting) | 1 | 2 | 59 | 3100 | 0.60 | 5 | 2 | 200 | Fertilizer spreader | N-P-K | 800.00 | 500 KP; 300N |
| - Mechanized transplantation | 1 | 1 | 73 | 3800 | 4.00 | 39 | 1 | 380 | Transplanter | Cuttings | n. 7000 | |
| - Chemical weeding post-planting | 1 | 1 | 59 | 3100 | 0.80 | 6 | 1 | 250 | Sprayer | Goal | 2.00 | |
| - Irrigation | 1 | 1 | 59 | 3100 | 7.00 | 56 | 1 | 300 | Pump and sprinkler | Water | 400,000 | |
| - Milling | 1-2-3-4-5-6-7-8-9-10-11-12-13-14-15-16 | 1 | 26 | 2035 | 6.00 | 30 | 2 | 380 | Milling machine | | | |
| - Harrowing | 1-2-3-4-5-6-7-8-9-10-11-12-13-14-15-16 | 1 | 80 | 4100 | 1.20 | 14 | 1 | 500 | Harrow | | | |
| - Stump grinding at the end cycle | 15 or 16 | 1 | 199 | 8700 | 8.75 | 340 | 1 | 500 | Stump grinder | | | |
| **Harvesting** | | | | | | | | | | | | |
| *-Option 1 (2y)—harvesting every 2 years* | | 1 | | | | | | | | | | |
| - Felling (tractor with disksaw) | | 1 | 59 | 3100 | 2.13 | 17 | 1 | 180 | Disk saw | | | |
| - Extraction (tractor with grapple) | 2-4-6-8-10-12-14-16 | 1 | 80 | 5500 | 2.84 | 37 | 1 | 150 | Log grapple | | | |
| - Chipping (farm chipper) | | 1 | 106 | 5500 | 10.22 | 195 | 1 | 1870 | Chipper | | | |
| - Moving and load (chipwood) | | 1 | 74.50 | 7130 | 8.76 | 98 | 1 | | | | | |
| *-Option 2 (2y)—harvesting every 2 years* | | 1 | | | | | | | | | | |
| - harvesting (forage harvester) | | 1 | 350 | 12,000 | 1.25 | 72 | 1 | | | | | |
| - Extraction (tractor with trailer) | 2-4-6-8-10-12-14-16 | 1 | 73 | 3800 | 1.25 | 14 | 2 | 600 | Trailer | | | |
| - Moving stored chipwood | | | 74.5 | 7130 | 2.84 | 35 | 1 | | | | | |
| - Load chipwood (biomass plant) | | 1 | 125 | 8 | 74.50 | 35 | 1 | | | | | |
| *-Option 1 (3y)—harvesting every 3 years* | | 1 | | | | | | | | | | |
| - Felling (tractor with disksaw) | | 1 | 59 | 3100 | 2.90 | 26 | 1 | 180 | Disk saw | | | |
| - Extraction (tractor with grapple) | 3-6-9-12-15 | 1 | 80 | 5500 | 3.76 | 50 | 1 | 150 | Log grapple | | | |
| - Chipping (farm chipper) | | 1 | 106 | 5500 | 13.94 | 244 | 1 | 1870 | Chipper | | | |
| - Moving and load (chipwood) | | | 74.5 | 7130 | 13.15 | 147 | 1 | | | | | |
| *-Option 2 (3y) harvesting every 3 years* | | 1 | | | | | | | | | | |
| - harvesting (forage harvester) | | 1 | 350 | 12,000 | 1.78 | 103 | 1 | | | | | |
| - Extraction (tractor with trailer) | 3-6-9-12-15 | 1 | 73 | 3800 | 1.78 | 19 | 2 | 600 | Trailer | | | |
| - Moving stored chipwood | | 1 | 74.5 | 7130 | 4.26 | 52 | 1 | | | | | |
| - Chipwood load | | 1 | 74.5 | 7130 | 13.15 | 147 | 1 | | | | | |
| *-Option 3 (4y)—harvesting every 4 years* | | 1 | | | | | | | | | | |
| - Felling (manual with chainsaw) | | 1 | 1.7 | 4 | 85.20 | 43 | 1 | | | | | |
| - Extraction (tractor winch) | 4-8-12-16 | 1 | 70 | 3800 | 24.34 | 281 | 1 | 330 | Winch | | | |
| - Chipping (farm chipper) | | 1 | 106 | 5500 | 17.53 | 307 | 1 | 1870 | Chipper | | | |
| - Moving and load chipwood | | 1 | 74.5 | 7130 | 17.53 | 196 | 1 | | | | | |

**Table 2.** *Cont.*

| Operation | Period (Years) | Machine | | | | | | Equipment | | Technical Input | | |
|---|---|---|---|---|---|---|---|---|---|---|---|---|
| | | Operation (n./ha) | Power (kW) | Weight (kg) | Work Time (h/ha) | Fuel (L/ha) | Machine (N.) | Weight (kg) | Type | Type | Quantity (kg) | Rates (kg/ha) |
| **-Option 4 (4y)—harvesting every 4 years** | | 1 | | | | | | | | | | |
| - Felling (shear head) | | 1 | 69 | 17,000 | 17.04 | 194 | 1 | 1350 | Shear head | | | |
| - Extraction (skidder) | 4-8-12-16 | 1 | 90 | 8000 | 5.68 | 84 | 1 | | | | | |
| - Chipping (farm chipper) | | 1 | 106 | 5500 | 17.53 | 307 | 1 | 330 | Chipper | | | |
| - Moving and load (chipwood) | | | 74.5 | 7130 | 17.53 | 196 | 1 | | | | | |
| **-Option 3 (harvesting every 5 years)** | | 1 | | | | | | | | | | |
| - Felling (manual with chainsaw) | | 1 | 1.7 | 4 | 88.75 | 45 | 1 | | | | | |
| - Extraction (tractor winch) | 5-10-15 | 1 | 70 | 3800 | 28.03 | 324 | 1 | | Winch | | | |
| - Chipping (farm chipper) | | 1 | 106 | 5500 | 20.72 | 362 | 1 | 330 | Chipper | | | |
| - Moving and load (chipwood) | | 1 | 74.5 | 7130 | 21.91 | 245 | 1 | | | | | |
| **-Option 4 (harvesting every 5 years)** | | | | | | | | | | | | |
| - Felling (shear head) | | 1 | 90 | 8000 | 19.36 | 220 | 1 | 1350 | Shear head | | | |
| - Extraction (skidder) | 5-10-15 | 1 | 90 | 8000 | 6.66 | 99 | 1 | | | | | |
| - Chipping (farm chipper) | | 1 | 106 | 5500 | 20.72 | 362 | 1 | | Chipper | | | |
| - Moving and load (chipwood) | | 1 | 74.5 | 7130 | 21.91 | 245 | 1 | | | | | |

For the construction of the inventory data, all inputs and outputs were collected and analyzed as primary and secondary data. The primary data were obtained directly from years of experimentation on the cultivation of SRC poplar. For some data not easily available, the database of the SimaPro 8.0.1 code, Ecoinvent 3 dataset (secondary data), was used. For each mechanical operation, the main technical characteristics were considered, such as the type of machine and equipment used, the engine power, the hours of work performed, the fuel and lubricant consumption, to evaluate the direct emissions of exhausted gases generated by the tractors, and the indirect emissions generated by the materials used for the construction of the agricultural machines used. The production processes were initially extrapolated from the SimaPro database and then modified, according to data collected directly in the field, only in the part of the tractors and equipment used and the consumption of diesel and lubricating oil, leaving unchanged the data relating to emissions into the air and onto the soil. All the operations necessary for the establishment of the plantations were considered and analyzed, as well as the post-planting management phases over the years, including the restoration of the field at the end of the cycle with grinding of the stumps [17].

Emissions related to the use of fertilizers and herbicides were determined based on the data available in the literature and of the outputs returned by the scientific software EFE-So (v 2.0.0.6; Fusi and Fusi) according to the model in [18]. $CO_2$ emissions from urea fertilization were calculated according to [19]. Herbicide emissions to air, surface water and groundwater were assessed using PestLCI 2.0 model [20]. A dry matter loss of 7% [21] was considered for wood harvesting systems that involved extracting the whole tree, drying at the landing site and chipping with a forest chipper when the moisture content of biomass reached 35%. Biomass storage was considered in the form of stacked and branchless trees. For these harvesting systems, 14.3 Mg per hectare per year of wood chips were produced. FBHS, on the other hand, considers the storage of fresh wood chips, with an average moisture content of 53%, in covered piles, which, during storage, are subjected to an average dry matter loss of 22% [22]. For the latter harvesting system, the quantity of wood chips obtainable from one hectare of poplar was 12 $Mg^{-1}$ $ha^{-1}$ $y^{-1}$ (35% M.C.) after storage. Considering the data reported by various authors on biomass production from poplar groves subjected to different cutting cycles [16,23–25], in Figure 2, a prudential estimate of the dry biomass production for the four cycles considered in the study is shown. The average biomass production at farm gate was assumed for all the cases examined to be equal to 10 Mg of dry matter per hectare, per year.

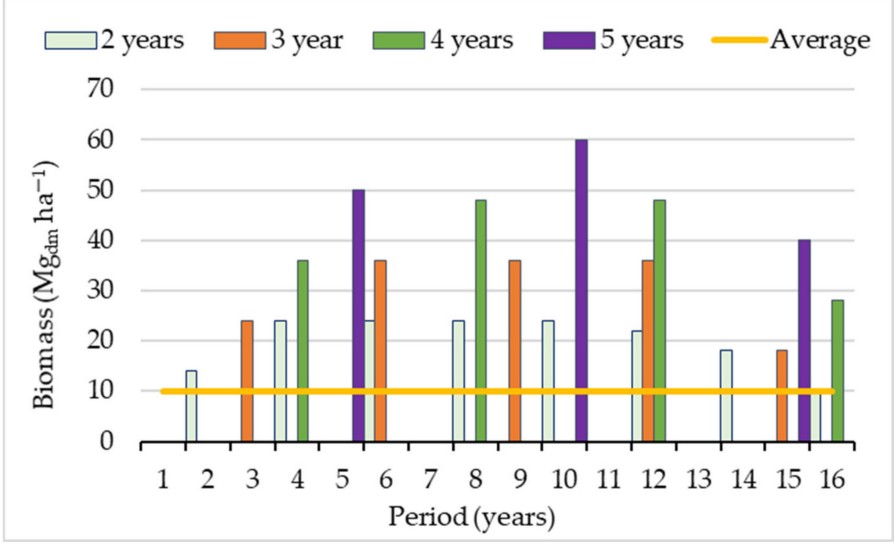

**Figure 2.** Estimate of dry biomass production in relation to the four cutting cycles considered.

Soil carbon (C) sequestration contributes to around 89% of the global mitigation potential of agriculture [26]. The amount of SOC stored in each soil is dependent on the balance between the amount of C entering the soil and the amount of C leaving the soil. The capacity of soils to store C is limited and the organic C content of soils tends towards equilibrium [27]. There is a high level of uncertainty regarding SOC estimation in soils, and there is no consensus or standard procedures on how to account for atmospheric carbon removals and releases [28–30]. According to the model reported by Whittaker et al. [31], an SRC plantation can store 8 Mg of stable organic carbon (SOC) at the end of the life cycle (corresponding to around 29 Mg $CO_2$ ha$^{-1}$). According to the model cited above, in the present research, the total C sequestered per hectare during the life of the SRC plantations studied, with an average yield of 10 Mg$_{dm}$ ha$^{-1}$ y$^{-1}$, would correspond to 7.61 Mg C ha$^{-1}$ (27.9 Mg $CO_2$ ha$^{-1}$). Although, in this study, SOC was not included in the calculation of the $CO_2$eq emitted, some considerations were subsequently reported to understand the positive impact provided by the SOC item in the final environmental performance results of the analyzed supply chains.

After considering all the agricultural phases, the impacts and resources of these phases (initially referred to a hectare of land) were compared to 1 GJ of biomass produced. This was possible by transforming the total production (Mg ha$^{-1}$) into energy (GJ ha$^{-1}$) since the low heating value (LHV) of poplar wood was calculated according to the Hartmann formula and considered equal to 11.2 MJ per kg of wood chips. The total inputs and emissions referring to one hectare were then divided by the production per hectare expressed in equivalent energy. In this way, it was possible to obtain the share of each agricultural phase to be attributed to 1 GJ of biomass produced. Average annual emissions and inputs were increased by the amount of inputs used and emissions generated over the years for planting, management, harvesting and removal, divided by the estimated life years of the crop (15 and 16 years). Reference was also made to an average annual production, calculated considering the yields obtained from poplar plantations during the years of their life cycle.

As the final phase, the results were evaluated, the weaknesses of the production phases were identified and the possibilities for improvement were defined.

## 3. Results and Discussion

The results in terms of emissions of kg $CO_2$eq per GJ of thermal energy produced are shown in Figure 3. The differences are highlighted above all between the two cases referring to the two-year cutting cycles compared to all the other cases. Despite the small number of observations without repetitions, which increases the margin of error, we still wanted to proceed with a statistical analysis by processing the data divided into four groups coinciding with the four cutting cycles. Welch's F test for unequal variance was performed and the results showed statistically significant differences ($p < 0.05$) between the first group (two-year cutting cycle) compared to the other three groups (cutting cycle of three, four and five years). From the first observations, it can be stated that more frequent harvesting operations contribute to increasing the number of agricultural practices adopted. This aspect is more evident especially in the case of the SRC plantations with a two-year cutting cycle (Figure 3), where the fertilization represented 49% of the overall emissions of the wood chip production.

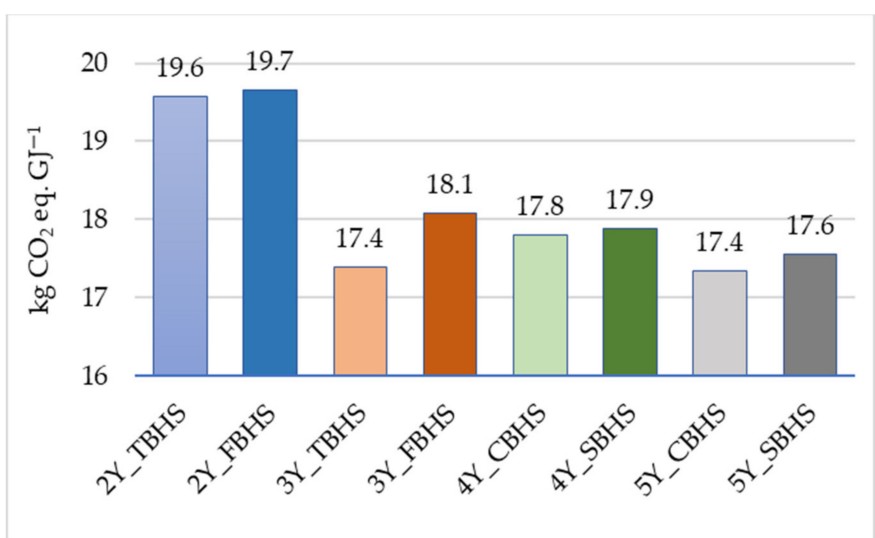

**Figure 3.** Emissions per GJ of thermal energy produced (in kg $CO_2$eq) with reference to the TBHS, FBHS, CBHS and SBHS harvesting systems (IPCC GWP 100a).

With specific reference to nitrogen fertilization (over 1 Mg of N in 16 years), it is responsible for 33% of the overall emissions referring to the production cycle of wood chips (34.6 kg $CO_2$eq per Mg of biomass at 35% of MC). A lower nitrogen intake was recorded in the five-year cutting cycle (around 0.4 Mg of N in 15 years), as it was administered only twice in correspondence with the first two harvests. This helped to reduce emissions attributable to fertilizers by nearly 60% compared to those of two-year cycles. The choice of the type of fertilizers to be used during the management of the plantations is therefore extremely important, as well as the application of good agricultural practices to ensure the maintenance of soil fertility, aimed at achieving a balance between the organic substance removed and the inputs of fertilizers, avoiding as much as possible the loss of nitrates by leaching. These results must be considered well-established when discussing emissions in agriculture [10,32–36]. In the analysis of the four different production cycles, slightly lower $CO_2$ emissions were recorded in reference to the use of a lower level of mechanization. More evident differences were recorded for the two-year production cycles (19.6–19.7 kg $CO_2$ $GJ^{-1}$), compared to all the other cases, which, on the other hand, had a lower impact in environmental terms (17.4-18.1 kg $CO_2$ $GJ^{-1}$).

According to the results of the study, the highest $CO_2$ emissions were therefore attributable to the two cases of the biennial poplar chain. In fact, for each GJ of thermal energy produced by the combustion of the biomass obtained from these cycles, a maximum of 19.7 kg $CO_2$eq was generated (Figure 3). However, within the two-year supply chain, the two cases examined were practically similar, showing differences contained within 1%.

This minimal difference is essentially attributable to two aspects, which, in the biennial poplar supply chains, are compensated: on the one hand, in the 2Y_FBHS case, there were greater storage losses for fresh wood chips, which led to greater emissions from the supply chain, while, on the other hand, in the 2Y_TBHS case, higher emissions were produced, mainly due to the increase in the number of operations to be carried out for the production of wood chips.

Figure 4 shows the $CO_2$ emissions per unit of dry biomass for the eight reference cases. Additionally, based on these values, within the limits already described above, a statistical analysis was performed with Welch's F test for unequal variance, returning a non-significant result ($p > 0.05$). The values referring to the four groups did not therefore show statistically significant differences between them. The $CO_2$ emissions related to the FBHS harvesting system (Figure 4) were due to the field harvesting phase of the forage harvester for 73% (18.8 kg$CO_2$eq $Mg_{dm}^{-1}$) and to the movement of wood chips in the piles for the remaining 27% (6.9 kg$CO_2$eq $Mg_{dm}^{-1}$). In the TBHS system, which involved the use of three

different machines, most of the emissions, equal to 78.2%, were attributable to the chipping operation performed with a forest chipper at the landing site (33 $kgCO_2eq\ Mg_{dm}^{-1}$). Regarding the remaining part, 14.4% was due to the extraction of whole trees with a tractor with a winch from the plantation to the landing site (6.1 $kgCO_2eq\ Mg_{dm}^{-1}$), and 7.4% to the cutting of the trees with the TBHS system (3.1 $kgCO_2eq\ Mg_{dm}^{-1}$).

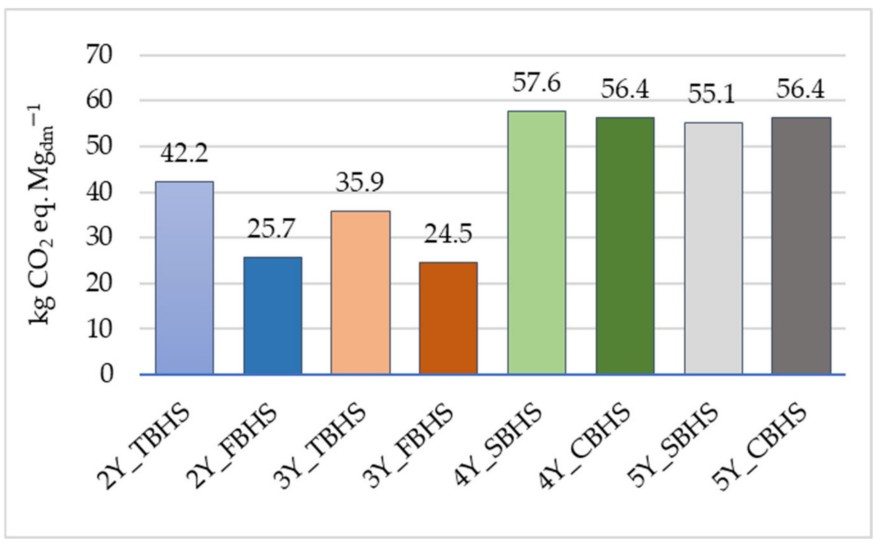

**Figure 4.** Emissions per $Mg_{dm}$ of biomass produced (in kg $CO_2eq$) with reference to the TBHS, FBHS, CBHS and SBHS harvesting systems, considering storage losses and excluding emissions from all other agricultural practices (IPCC GWP 100a).

In the FBHS scenario, much higher dry matter losses were found than in the TBHS scenario, due to the different storage system used. An optimization of this phase could lead the system to a reduction in emissions, making it more competitive with respect to the TBHS. This effect can also be applied in the case of three-year cutting cycle that applies the same harvesting logistics as the two-year one, even if, here, the differences to be bridged were slightly more marked (4%). In this case, therefore, it would be more difficult to obtain a possible reduction in emissions to the level of those recorded for the TBHS. In general, cutting shifts longer than two years showed lower emissions. Figure 3 shows that the two supply chains generated a very similar quantity of $CO_2$ in terms of GJ produced (an average of 17.6 kg $CO_2eq\ GJ^{-1}$) because the harvesting methods of the four-year and five-year cycles were the same. The harvesting systems used in four-year and five-year cycles produced, on average, 43% and 124% more $CO_2$ than the TBHS and FBHS systems, respectively, both used in the two-year and three-year cycles. From an environmental point of view, the results of the study show that the emissions of greenhouse gases produced by the analyzed wood-energy supply chains range from a maximum of 19.7 (biennial supply chain) to a minimum of 17.4 kg $CO_2eq$ per GJ of thermal energy produced by the biomass boiler considered. This result, although higher than that reported in other studies [25,37–41], is still well below the $CO_2$ emissions emitted by a boiler of the same size fueled by fossil fuels.

In Figure 5, it can be seen how the transition from a diesel boiler to a biomass-fueled biennial poplar wood chip (which was the least efficient compared to the other scenarios analyzed) allows a 77% reduction in greenhouse gas emissions. In the study, $CO_2$ stored as stable soil organic carbon (SOC) was not considered in the calculation, although the literature reports various environmental studies in which the immobilized carbon in the soil is considered [42–47]. If, in the present study, the estimated 7.61 Mg C $ha^{-1}$ (corresponding to 27.9 Mg $CO_2\ ha^{-1}$) accumulated at the end of the cycle is included, then the greenhouse gas emissions produced to generate 1 GJ of thermal energy would be even lower. In fact, considering a life cycle of 16 years and a production amount of 130 GJ, 13.4 kg $CO_2$ would

have been saved per GJ produced. In this case, the 2Y_TBHS scenario reported in Figure 4 would result in the emission of only 6.3 kg $CO_2$eq per thermal GJ generated.

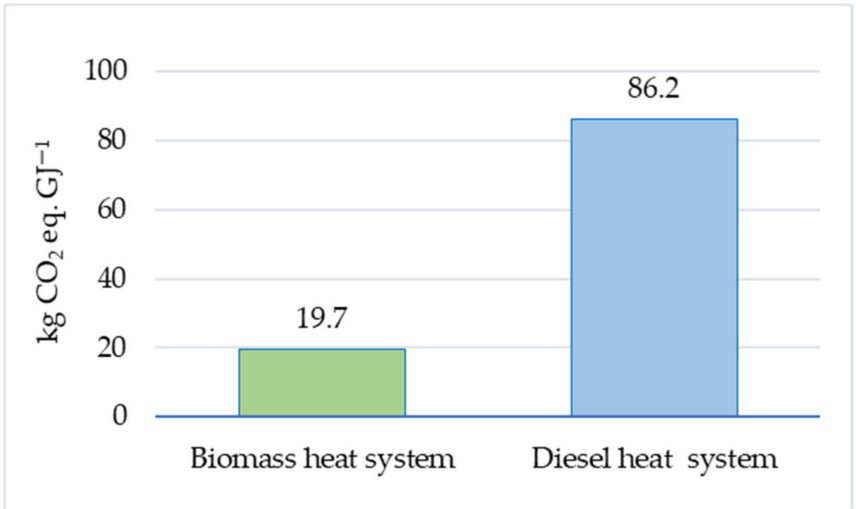

**Figure 5.** Comparison between the emissions (in kg $CO_2$eq) generated by a biomass boiler connected to the less efficient biennial poplar supply chain and a diesel boiler to produce 1 GJ of thermal energy.

## 4. Conclusions

The use of fossil fuels for energy production has led over time to high greenhouse gas emissions. This is why we are seeing increasing interest in renewable energy. This includes biomass from short-rotation agroforestry crops such as poplar due its energy potential, as an alternative to fossil fuels and the consequent production of GHG.

With the aim of evaluating the environmental sustainability of small-scale self-consumption wood-energy chains based on poplar SRC for heat generation, this study focused on the carbon footprint of various crop management scenarios represented by different harvesting systems and cutting cycles.

The energy production obtained from the poplar three-year and five-year cutting cycles (TBHS and CBHS harvesting systems) was found to be the most sustainable, but the most evident differences were highlighted only between the two-year supply chains and those of the other cutting cycles. Several studies have highlighted how the type and quantity of fertilizer applied is important in environmental performance and, in this context, the optimization of the inputs used becomes fundamental.

Clearly, the most sustainable harvesting method is characterized by fewer production steps. In our case, the FBHS harvesting system, characterized by chipping plants in the field, was the most sustainable compared to the two-stage harvesting systems (TBHS, CBHS and SBHS). However, it should be noted that the FBHS system results in higher dry matter losses during the storage phase and that the benefit in terms of emissions is not very evident when considering the entire energy chain.

It is therefore essential to optimize the storage of fresh wood chips to reduce dry matter losses and thus obtain a further reduction in total emissions from the energy supply chain.

Analyzing our results, we can affirm that producing thermal energy from biomass, compared to that obtained from fossil fuel, allows a reduction in greenhouse gas emissions equal to 77%. A further improvement (93% reduction of emissions) is possible if we consider the $CO_2$ stored in the soil in the form of SOC at the end of SRC life cycle. The stabilization of $CO_2$ in soil as soil organic carbon should be further investigated as this is the aspect that makes a bioenergy supply chain more sustainable.

**Author Contributions:** Conceptualization, G.S.; methodology, G.S., A.S.; software, A.S.; formal analysis, G.S., A.S.; data curation, G.S., A.S.; writing—original draft preparation, G.S., A.S., A.A.; writing—review and editing, G.S., A.S., A.A., V.C. All authors have read and agreed to the published version of the manuscript.

**Funding:** This research was funded by the Italian Ministry of Agriculture, Food and Forestry Policies (MiPAAF), grant D.D. n. 26329, 1 April 2016, project AGROENER "Energia dall'agricoltura: innovazioni sostenibili per la bioeconomia".

**Institutional Review Board Statement:** Not applicable.

**Informed Consent Statement:** Not applicable.

**Data Availability Statement:** Data is contained within the article.

**Conflicts of Interest:** The authors declare no conflict of interest.

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
