# Peer review of "Environmental Sustainability of Heat Produced by Poplar Short-Rotation Coppice (SRC) Woody Biomass"

_forests, doi:10.3390/f12070878_

Round 1

Reviewer 1 Report

In the abstract in line 11 the abbreviation SRC has to be explained (this can also be done by adding "Environmental sustainability of thermal energy produced by poplar short rotation coppice (SRC) woody biomass." to the title). The authors adopted this way for the abbreviation LCA - line 44 where it was explained when introducing the concept of Life Cycle Assessment (LCA). 

Table 1: Why is a moisture content of 35% assumed when the recommendation for firewood is 15-20% due to the formation of toxic substances? (EN 13556:2003, ISO 18125). 
Water content in diesel fuel is low but not zero (this is determined by the legislation of individual countries - an example of an article relating to the issue:
Macioszek, Ł.; Rybski R.; Low-Frequency Dielectric Spectroscopy Approach to Water Content in Winter Premium Diesel Fuel Assessment, De Gruyter Oldenbourg, 2017, https://doi.org/10.1515/9783110449822-008

I propose to perform a significance of differences test to enrich the results and indicate how significant differences give the results. 

Author Response

Point 1:

In the abstract in line 11 the abbreviation SRC has to be explained (this can also be done by adding "Environmental sustainability of thermal energy produced by poplar short rotation coppice (SRC) woody biomass." to the title). The authors adopted this way for the abbreviation LCA - line 44 where it was explained when introducing the concept of Life Cycle Assessment (LCA).

Response 1:

As suggested, the title of the manuscript was changed by adding "(SRC)" after "short rotation coppice".

Point 2:

Table 1: Why is a moisture content of 35% assumed when the recommendation for firewood is 15-20% due to the formation of toxic substances? (EN 13556:2003, ISO 18125).

Water content in diesel fuel is low but not zero (this is determined by the legislation of individual countries - an example of an article relating to the issue:

Macioszek, Ł.; Rybski R.; Low-Frequency Dielectric Spectroscopy Approach to Water Content in Winter Premium Diesel Fuel Assessment, De Gruyter Oldenbourg, 2017, https://doi.org/10.1515/9783110449822-008.

Response 2:

Thank you for your remark. A biomass water content of 35% was reported because this is the level that guarantees a good functioning of the biomass power plant. This is in relation to the technical characteristics of the plant which is equipped with a mobile grid capable of effectively burning biomass with water content even higher than that indicated in the text of the manuscript. No reference is made to compliance with the regulatory limits of water content relating to the commercial product, as the wood chips are produced and transformed into thermal energy within the farm.

As for the water content of the diesel, it was not considered since, as also affirmed by the reviewer, its value is generally close to 0. However, accepting the suggestion, in Table 1 was added the value "≤0.05%", which represents the limit water content imposed by the legislation in force in Italy.

Point 3:

I propose to perform a significance of differences test to enrich the results and indicate how significant differences give the results.

Response 3:

The research analysed 8 different case studies, for this reason no repetitions are available. Under these conditions, a statistical analysis can be subject to a high possibility of error. However, accepting the suggestion, a statistical analysis was performed using the Welch F test for unequal variance. The results have been reported in the text, with reference to the comments on the data in Figures 3 (line 215) and Figure 4 (line 255).

Reviewer 2 Report

This paper talks about a relevant aspect in the world debate of reducing the climate-altering gases into the atmosphere using short rotation coppices plantations. The results obtained in this article are significant for the environmental impact of 1 GJ of heat energy produced by the various forest rotation 16 plants considering the entire life cycle, from the field stage to the thermal energy pro-17 duction.
In the Abstract it is introduced the term  IPCC method without any description. Please rectify this aspect.
The Introduction is very well described and organized with the overall and the authors' methods to use in the article. However, the authors can talk a little about the short rotation coppice before the application to poplar.
In results and discussion, the authors said, "The various scenarios examined did not reveal significant differences," but 2Y_TBHS and 2Y_FBHS are different from others. Can you explain?

Author Response

Point 1:

In the Abstract it is introduced the term IPCC method without any description. Please rectify this aspect.

Response 1:

As suggested, a sentence has been added in "Materials and Methods" to make the citation clearer (line 132).

Point 2:

The Introduction is very well described and organized with the overall and the authors' methods to use in the article. However, the authors can talk a little about the short rotation coppice before the application to poplar.

Response 2:

As suggested, an explanatory sentence has been inserted in line 47.

Point 3:

In results and discussion, the authors said, "The various scenarios examined did not reveal significant differences," but 2Y_TBHS and 2Y_FBHS are different from others. Can you explain?

Response 3:

According to your observation, the text was changed and clarified, and a sentence added to explain the differences (line 213)